# Naa80 is required for actin N-terminal acetylation and normal hearing in zebrafish

Rasmus Ree[1], Sheng-Jia Lin[2], Lars Ole Sti Dahl[1], Kevin Huang[2], Cassidy Petree[2], Gaurav K Varshney[2], Thomas Arnesen[1,3]

**Actin is a critical component of the eukaryotic cytoskeleton. In animals, actins undergo unique N-terminal processing by dedicated enzymes resulting in mature acidic and acetylated forms. The final step, N-terminal acetylation, is catalyzed by NAA80 in humans. N-terminal acetylation of actin is crucial for maintaining normal cytoskeletal dynamics and cell motility in human cell lines. However, the physiological impact of actin N-terminal acetylation remains to be fully understood. We developed a zebrafish *naa80* knockout model and demonstrated that Naa80 acetylates both muscle and non-muscle actins in vivo. Assays with purified Naa80 revealed a preference for acetylating actin N-termini. Zebrafish lacking actin N-terminal acetylation exhibited normal development, morphology, and behavior. In contrast, humans with pathogenic actin variants can present with hypotonia and hearing impairment. Whereas zebrafish lacking *naa80* showed no obvious muscle defects or abnormalities, we observed abnormal inner ear development, small otoliths, and impaired response to sound. In conclusion, we have established that zebrafish Naa80 N-terminally acetylates actins in vitro and in vivo, and that actin N-terminal acetylation is essential for normal hearing.**

## Introduction

Actin, the most abundant protein in animal cells, plays a crucial role in a wide range of cellular functions. Actin filaments not only maintain cell shape and rigidity but also provide the scaffold for myosin-driven movements, essential for processes like protrusion formation and muscle contraction (Pollard & Cooper, 2009; Svitkina, 2018). These actin–myosin powered cell movements are widespread and particularly important during development (see, e.g., [Bonneau et al, 2011]). Actin function is regulated by various mechanisms, including posttranslational modifications (PTMs) and interactions with actin-binding proteins (ABPs) (Varland et al, 2019). In

animals, actin undergoes a unique N-terminal (Nt) maturation process culminates in Nt-acetylation (Vandekerckhove & Weber, 1978; Rubenstein & Martin, 1983a, 1983b). The enzyme responsible for this final maturation step, NAA80, was identified only recently (Drazic et al, 2018; Goris et al, 2018; Wiame et al, 2018). Nt-acetylation of cytoplasmic actin isoforms (β-actin and γ-actin in humans) is highly prevalent, with nearly 100% of actin modified in this way in HAP1 and other mammalian cells (Drazic et al, 2018). This acetylation is absent in *NAA80*-KO cells (Drazic et al, 2018, 2022). Although non-acetylated actin can still polymerize and contribute to functional F-actin networks, it does so more slowly in vitro, both in terms of polymerization and depolymerization. This leads to an increased filamentous to globular (G/F) actin ratio compared with WT cells (Drazic et al, 2018). *NAA80*-KO cells exhibit several phenotypic changes including increased rates of migration, increased cell size, Golgi fragmentation, and increased numbers of filopodia and lamellipodia (Aksnes et al, 2018; Drazic et al, 2018; Beigl et al, 2020). NAA80 possesses a conserved polyproline loop crucial for its interaction with the actin-binding protein PFN2 (Rebowski et al, 2020; Ree et al, 2020). This NAA80-PFN2 association facilitates efficient posttranslational Nt-acetylation of G-actin monomers before they are incorporated into F-actin.

The first description of patients with a pathogenic *NAA80* variant was published recently (Muffels et al, 2021). The two siblings presented with progressive high-frequency hearing loss, abnormal craniofacial features, developmental delay, and mild muscle weakness in both the trunk and the limbs. Genetic sequencing revealed the patients were homozygous for a c.389T>C (Leu130Pro) variant in *NAA80* gene which led to instability of the NAA80 protein and reduced actin Nt-acetylation to about half the level observed in control cells. Several of the observed symptoms were similar to those associated with variants that impair actin function.

Whereas the phenotype of NAA80 knockout in single cells is well characterized, no animal models currently exist that explore the complete loss of *NAA80* function. To address this gap, we used zebrafish (*Danio rerio*) to investigate the impact of *naa80* knockout and characterize the substrate specificity of Naa80. Our data reveal that zebrafish Naa80 is responsible for Nt-acetylating actin-type

[1]Department of Biomedicine, University of Bergen, Bergen, Norway   [2]Genes and Human Disease Research Program, Oklahoma Medical Research Foundation, Oklahoma City, OK, USA   [3]Department of Surgery, Haukeland University Hospital, Bergen, Norway

Correspondence: rasm@norceresearch.no; gaurav-varshney@omrf.org; thomas.arnesen@uib.no
Rasmus Ree's present address is NORCE Climate and Environment—NORCE Norwegian Research Centre, Bergen, Norway
Lars Ole Sti Dahl's present address is Virology Unit, Faculty of Veterinary Medicine, Norwegian University of Life Sciences, Ås, Norway

N-termini and zebrafish lacking Naa80 (*naa80 −/−*) show a complete absence of Nt-acetylation in both cytoplasmic and muscle actins from skeletal and cardiac muscle. Interestingly, despite this lack of actin N-acetylation, *naa80 −/−* zebrafish appear morphologically normal and exhibit no apparent developmental issues. However, we observed that one of the knockout lines failed to produce eggs, suggesting a potential role of actin Nt-acetylation in egg-laying processes. In addition, *naa80* KO larvae displayed smaller otoliths, reduced hair cell bundles, decreased hair cell viability in the inner ear. These inner ear abnormalities were associated with a decreased startle response, indicating a potential hearing impairment.

# Results

## A *D. rerio* Naa80 ortholog displays Nt-acetyltransferase activity towards acidic, actin-type N-terminal peptides in vitro

The zebrafish, like other animals, possesses several actins that vary in their Nt-sequences. The actin Nt-sequences have subtle differences but are invariably acidic (Fig 1A). Class I actins, including cytoplasmic 1 and cytoplasmic 2, have a stretch of three acidic amino acids at the N-terminus following the initiator methionine (iMet), while class II actins (alpha-cardiac 1a and 1b, alpha-skeletal 1a and 1b, and alpha-smooth) have iMet and cysteine residues N-terminal to 3–4 acidic residues.

The class I actins are presumed to be co-translationally acetylated by NatB (Van Damme et al, 2012) before the Nt-Ac-Met is removed by actin methionine aminopeptidase, the recently discovered actin maturation protease (Haahr et al, 2022) (Fig 1A). This reveals the neo-N-terminus at Asp2/Glu2, which then undergoes post-translational Nt-acetylation by NAA80 (Drazic et al, 2018). The class II actins (Met-Cys-Asp/Glu-…) are presumed to be substrates of MetAP1/2, removing the iMet and exposing the neo-N-terminus at Cys2 (Moerschell et al, 1990). This is then potentially Nt-acetylated by NatA (co-translationally [Heathcote et al, 2024]) before the Cys2 is removed by actin methionine aminopeptidase (Haahr et al, 2022), exposing the Nt-amine group of acidic Asp3/Glu3 as the acceptor of NAA80-mediated acetylation (Arnesen & Aksnes, 2023).

To identify the zebrafish ortholog of NAA80, we performed a protein BLAST (Altschul et al, 1990) search using the human NAA80 protein sequence as a reference. The top result, annotated in Uniprot (E7FBQ5), matches the N-alpha-acetyltransferase 80, NatH catalytic subunit due to its strong sequence similarity with NAA80 across different species (Goris et al, 2018). There are two exons (137 and 930 bp) and one 3,108 bp intron in this zebrafish gene. The two last bases of the first exon forms part of the start codon of the protein-coding region, which is almost completely contained in the longer second exon. Both the 5del/1in and 13del mutations are located within the second exon (Fig S1A). We cloned the cDNA encoding this protein (Refseq: XM_005167127.3) and expressed it in *Escherichia coli* BL21* cells, tagged it with maltose-binding protein (MBP) and His-tags for purification. After a two-stage purification process, we achieved an apparently homogenous MBP-E7FBQ5/zNaa80. We used the purified zNaa80 in an acetylation assay to determine its activity and substrate preference. The enzyme was

tested against peptides with different N-terminal sequences, and the activity readout is disintegrations per minute (DPM) (Drazic & Arnesen, 2017) normalized to the peptide with the highest activity (Fig 1B and C).

Initially, we assessed several peptides representing different NAT substrate classes (Fig 1B). NAA80 is characterized by its activity towards acidic N-termini without the initiator methionine, illustrated by the N-terminus DDDIA (from human β-actin), as well as EEED, DDEE, and DEDE, which are derived from human alpha-smooth muscle, alpha-cardiac muscle, and alpha-skeletal muscle, respectively. The peptide SESS is derived from a NatA substrate (Arnesen et al, 2009), MELL from a classical NatB substrate (Van Damme et al, 2012), MLGP from a NatC/NatE substrate (Evjenth et al, 2009), and MAPL from a NatF substrate (Van Damme et al, 2011b). The activity profiles of E7FBQ5, which favors acidic actin N-terminal peptides, strongly supports the identification of this protein as the zebrafish Naa80 ortholog.

## Dynamic expression of *naa80* mRNA at different developmental stages and adult tissues

To gain insight into the possible function of *naa80*, we performed whole-mount in situ hybridization (WISH) (Fig 2A–D) and RT-qPCR (Fig 2E) at various developmental stages to reveal the spatial and temporal *naa80* mRNA expression pattern. Notably, *naa80* expression was detected as early as 1-h post-fertilization (hpf) (Fig 2A), indicating maternal distribution of *naa80* mRNA. By 24 hpf, *naa80* mRNA exhibited ubiquitous expression, including the central nervous system, eyes, otic vesicles, and trunk muscles (Fig 2B and B′). Subsequently, at 48 and 72 hpf, *naa80* expression became restricted to the brain, eyes, and pectoral fin bud (Fig 2C, C′, and D). The RT-qPCR results further supported the presence of *naa80* expression throughout embryonic development (Fig 2E).

Furthermore, we performed RT-qPCR analysis on various tissues from one-year-old adult animal. The results revealed *naa80* is expressed in most tissues, with significant enrichment in the kidney, stomach, oocytes, testis, brain, and skeletal muscles (Fig 2F). This enrichment suggests a potential role for *naa80* in these tissues. In summary, *naa80* mRNA shows ubiquitous expression during embryonic development, gradually becoming more restricted to the brain, and it exhibits relatively higher levels in kidney and reproductive tissues during adulthood.

## *naa80 −/−* zebrafish are free from gross morphological phenotypes and develop normally

To investigate the in vivo function of zNaa80, we used CRISPR-Cas9 and a gRNA targeting the putative *naa80* gene (Fig 3A). We thus established two zebrafish knockout lines, one carrying an allele with a 5 basepair deletion and 1 basepair insertion (GCAGC>A, 5del/1in), and one carrying a 13 basepair deletion (13del) (Fig S1B). These mutations lead to a frameshift and missense mutations starting at amino acid residue 35–37, and premature stop codons after positions 85–88 (Figs 3B and S2). Neither predicted mutant protein variant contains the Ac-CoA binding motif, they are both nearly completely missing the GNAT fold, as well as the polyproline-rich

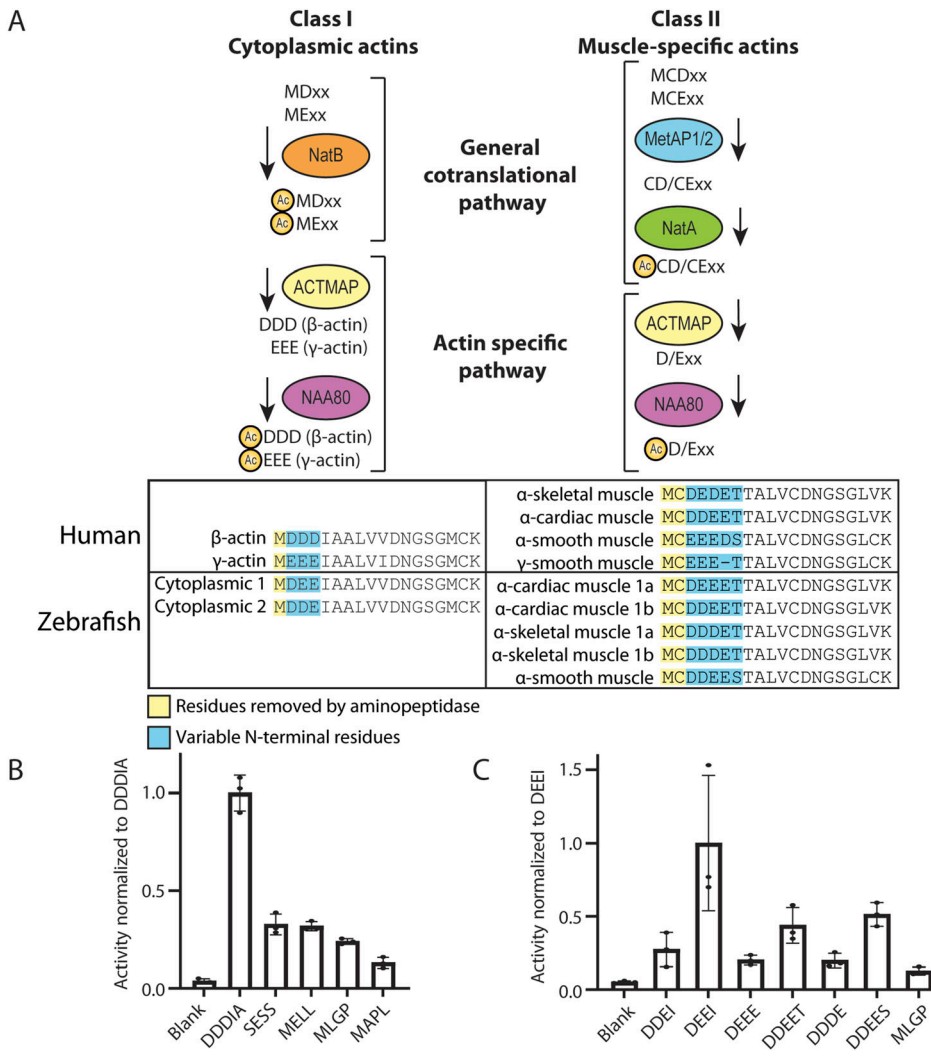

**Figure 1.  Actin N-terminal maturation scheme and the in vitro acetyltransferase activity of *Danio rerio* E7FBQ5/Naa80 towards actin-type N-termini.**
**(A)** Overview of actin N-terminal processing leading up to Nt-acetylation. Class I actins are first co-translationally Nt-acetylated by NatB, and then actin methionine aminopeptidase removes the acetylated Met, before NAA80 acts on the acidic neo-N-terminus. Class II actins are processed by MetAP1/2 before Nt-acetylation by NatA and removal of an acetylated Cys by actin methionine aminopeptidase before Nt-acetylation by NAA80. In both humans and zebrafish, actin is conserved with most of the amino acid differences located to the N-terminus. **(B)** Acetylation assays with purified MBP- E7FBQ5/zNaa80 and different peptides derived from human proteins. **(C)** Acetylation assays with purified MBP- E7FBQ5/zNaa80 and different peptides derived from zebrafish actins.

loop near the C-terminus. For these reasons, we expected the mutant Naa80 proteins to lack N-terminal acetyltransferase (NAT) activity. The knockout alleles were verified by fin clipping followed by PCR using *naa80*-specific primers followed by Sanger sequencing (for either allele; Fig S1C) or agarose gel electrophoresis for the 13del allele (Fig 3C).

These knockout fish displayed no obvious morphological abnormalities and developed normally. Body size was unaffected (Fig 4). Male *naa80* 13del+/− fish had an average body length of 3.447 cm, compared with 3.793 cm for WT fish which is a significant reduction ($P < 0.05$), However, *naa80* 13del−/− fish were 3.633 cm on average, suggesting that the *naa80* genotype is not likely to be the underlying cause. In all other aspects, body weight and length showed no specific morphological abnormalities in *naa80* +/− or −/− fish.

We attempted to obtain *naa80*−/− eggs by crossing *naa80* 5del/1in −/− and *naa80* 13del −/− fish in the F1 generation, but this did not yield any eggs. Consequently, the F2 generation was produced by incrossing *naa80*+/− fish and genotyping the

mature offspring. In the F2 generation, systematic mating tests were conducted to determine if this was a characteristic of the the *naa80*−/− phenotype. Multiple incrosses of *naa80* 5 bp del/1 bp ins −/− fish failed to yield eggs. However, *naa80* 13 bp del ± females crossed with *naa80* 13 bp del −/− males resulted in viable embryos with no observable phenotypic differences. Similarly, Spotty WT females mated with *naa80* 5del/1in −/− males produced viable embryos. On the other hand, *naa80* 5 bp del/1 bp ins −/− females crossed with Spotty wild-type (SWT) males produced no eggs.

Interestingly, incrossing *naa80* 13 bp del −/− fish in F3 generation produced several eggs that developed normally throughout the larval stage. This finding suggests that that neither maternal *naa80* nor maternal Nt-acetylated actin are necessary for normal development pre-mid blastula transition. Although we cannot definitively state that *naa80* −/− females are infertile, their fertility or egg-laying capability may be compromised, as we were unable to obtain eggs from *naa80* 5 bp del/1 bp ins −/− females.

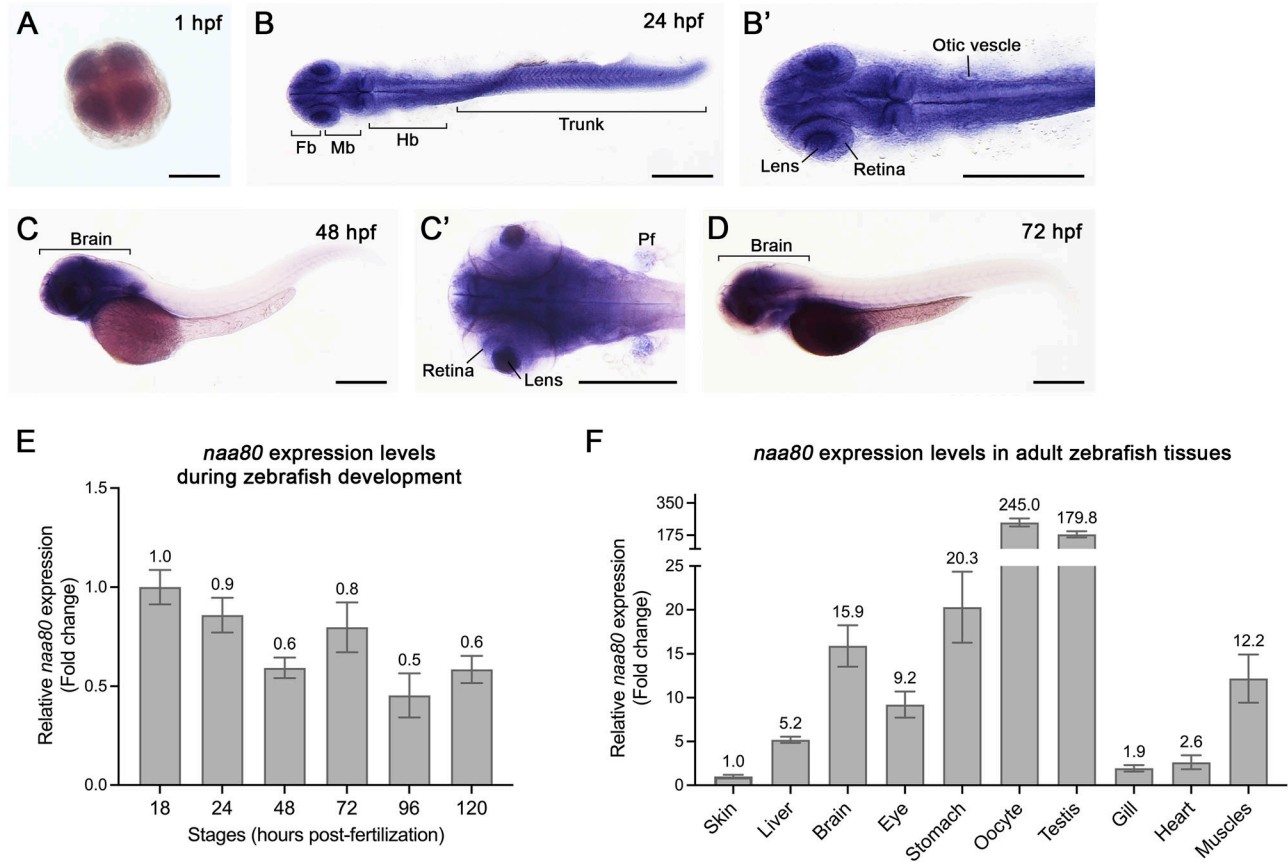

**Figure 2. The spatiotemporal expression of *naa80* mRNA by whole-mount in situ hybridization and RT-qPCR.**
**(A, B, B', C, C', D, E, F)** Whole-mount in situ hybridization was performed at different developmental stages of zebrafish embryos at 1 (A), 24 (B, B'), 48 (C, C') and 72 (D) hpf. RT-qPCR at different embryonic stages (E) and adult tissues (F). For (A, B, B', C'), dorsal view and anterior to the left. For (C, D), lateral view and anterior to the left. Fb, forebrain. Mb, midbrain. Hb, hindbrain. Pf, pectoral fin bud. Scale bar = 0.3 mm. For (E, F), each timepoint included biological triplicates as well as technical triplicates. **(E, F)** The expression levels were first normalized to the *18S* housekeeping gene and the expression levels were compared with 18 hpf embryos (E) or compared with skin (F). Data shown are mean ± SD and mean values were shown on each bar.

## Impaired N-terminal acetylation of cytoplasmic and muscle actins in *naa80 −/−* zebrafish

To assess the Nt-acetylation status of actins in *naa80* KO zebrafish, we dissected and lysed skeletal muscle and heart tissue. The soluble proteins were trypsinized and analyzed using LC/MS. Most *naa80 −/−* samples showed no Nt-acetylated actins; however, a small amount of Nt-acetylated cytoplasmic actin 2 (Ac-DDEI...) was detected in one of the *naa80* 5del/1in −/− heart samples (Fig 5, Tables S1 and S2). This could be residual Nt-acetylation by Naa10 or another NAT (Van Damme et al, 2011a; Ree et al, 2015), or it could result from carryover or contamination from a different sample. Nevertheless, nearly all Nt-acetylation appears to be absent in *naa80 −/−* fish.

In the *naa80 +/+* and *+/−* samples, we identified Nt-acetylated actin N-termini and very few to no unacetylated actin N-termini. This finding aligns with previous studies showing a high degree of actin Nt-acetylation (Drazic et al, 2018, 2022). From this, we deduced that one functional allele of *naa80* is sufficient for complete actin Nt-acetylation. In contrast, in the *naa80 −/−* samples from both knockout lines, we found little to no

detectable Nt-acetylated actin (Fig 5). Instead, we observed unacetylated actin N-termini, as well as a comparable number of actin nonterminal peptides in *naa80 −/−* samples, indicating that this is not due to underrepresentation of actins in the analyzed proteome (Table S1, listing all modified peptides). Other proteins may have been similarly affected by *naa80* knockout, showing Nt-Ac N-termini in the WT and *naa80 +/−* samples and free N-termini in *naa80 −/−* samples. We expected substrates of Naa80 to be Nt-acetylated in the WT and heterozygous samples and non-acetylated in the knockout samples. To test this, we compared the N-terminal peptides which had been found in both Nt-acetylated and Nt-free forms throughout the dataset (Table S3). We found 33 N-termini with both acetylated and non-acetylated forms, including the seven actin N-termini. Of these, actins were the only N-termini with acetylation status varying with *naa80* genotype. We thus found no evidence that other proteins than the actins were affected in their Nt-acetylation status by *naa80* knockout. We conclude that in skeletal and cardiac muscle, Naa80 is the enzyme responsible for class I and class II actin Nt-acetylation in zebrafish, as it is in human cells.

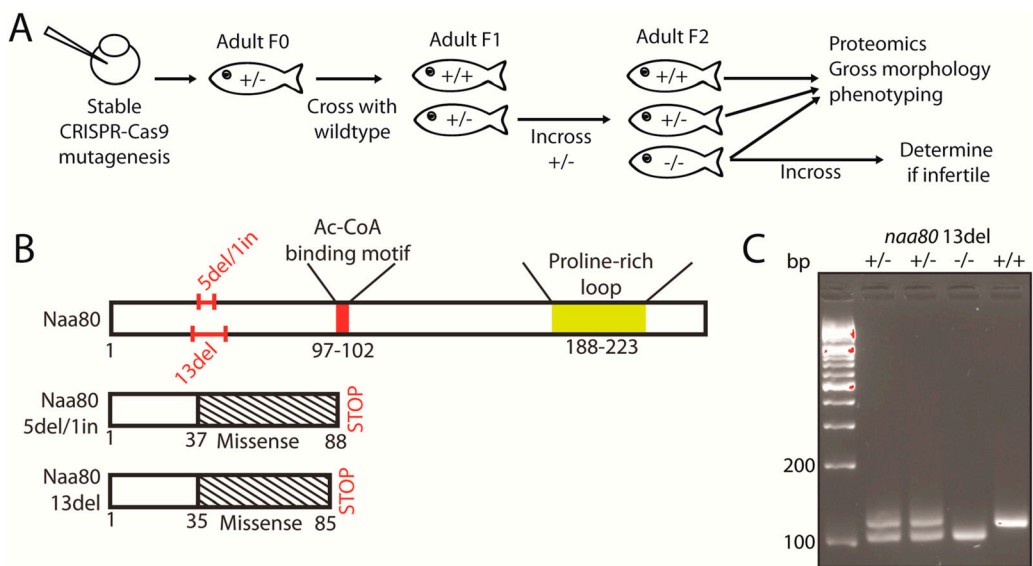

**Figure 3. Generation of stable mutant *naa80* knockout lines.**
**(A)** Overview of *naa80* mutant zebrafish crossing and experiments performed. **(B)** The Naa80 protein with location of 5del/1in and 13del mutations, as well as locations of the Ac-CoA binding motif (red) and proline-rich loop (yellow) highlighted. Also highlighted are the sense and missense regions and the premature stop codons which arise in the mutant alleles. **(C)** Representative PCR products (112 bp from WT allele, 99 bp from 13del allele) from *naa80* 13del fin clips visualized in agarose gel (4% agarose in lithium borate acetate buffer and GelRed).

### *naa80* F0 knockout (KO) zebrafish displayed hearing-related defects

A previous study reported that humans with biallelic missense variants in *NAA80* show decreased actin Nt-acetylation and increased polymerized actin, potentially contributing to high-frequency hearing loss (Muffels et al, 2021). However, no direct hearing-related assessments have been conducted in genetic mutant models. The specific mechanism by which aberrantly acetylated actin leads to the hearing loss phenotype in vivo remains unclear. In zebrafish, the sensory patches in the inner ear are the maculae and the cristae (Nicolson, 2005). The maculae are located in the otolith organs (utricle and saccule), while the cristae are located in the semicircular canals. Cristae detect head movement and help maintain balance by bending sensory hair cells in response to fluid movement. Maculae sense linear acceleration and gravity by detecting shifts in otoliths that bend the stereocilia of hair cells. The hair cells located in the inner ear's three cristae and two maculae, and the lateral line neuromast of zebrafish larvae provide an excellent model for investigating hearing mechanisms because of their functional similarities with human inner ear hair cells and are easily accessible for assessment (Elepfandt, 1988; Vona et al, 2020) (Fig 6A). Actin is crucial for the development and function of the inner ear hair cell, including shaping the hair cell, developing stereocilia bundles, and transporting vesicles.

To determine whether the loss of Naa80 affects hearing function and to address the fertility challenges in stable mutant lines, we used the transient CRISPR method, which efficiently generates null mutants in the founder generation (referred to as knockout or KO hereafter to distinguish them from stable mutants) (Lin et al, 2021). Similar to stable genetic *naa80* mutants, *naa80* knockouts did not

display any apparent gross morphological phenotypes. However, we observed a reduction in otolith size compared with controls following Cas9 protein injection (Fig 6B and C'). RT-qPCR showed a down-regulation of *naa80* expression in F0 KO (Fig 6D). Immuno-histochemistry using acetylated tubulin (Ac-tub, for kinocilia staining) and phalloidin (staining F-actin, for stereocilia staining) (Fig 6A) revealed that KOs exhibited a fewer hair cell bundles in the lateral crista compared with controls (Fig 6E–F"), and both number of hair cells per crista as well as stereocilia length reduced (Fig 6G and H). Moreover, the KOs demonstrated a reduced acoustic startle response compared with controls (Fig 6I), suggesting a hearing impairment. Reduced hair cell viability in KOs was also observed through the Yo-Pro-1 uptake experiment (Fig 6J–L). In summary, the findings of reduced otolith size, fewer hair cell bundles in the lateral crista of the inner ear, and viable hair cells in the lateral line neuromast indicating hearing impairment in *naa80* KOs. The analysis of acoustic startle responses further support these conclusions.

## Discussion

N-terminal acetylation is likely the most common protein modification in eukaryotes, with 50–90% of eukaryotic proteomes being Nt-acetylated (Arnesen et al, 2009; Goetze et al, 2009; Bienvenut et al, 2012). The functional impact of Nt-acetylation can vary from protein to protein and include protein folding, stability, degradation, complex formation, and subcellular targeting (Ree et al, 2018; Aksnes et al, 2019). Recent global analyses uncovered that protection from protein degradation is a major function of protein Nt-acetylation in multicellular eukaryotes (Mueller et al, 2021; Linster

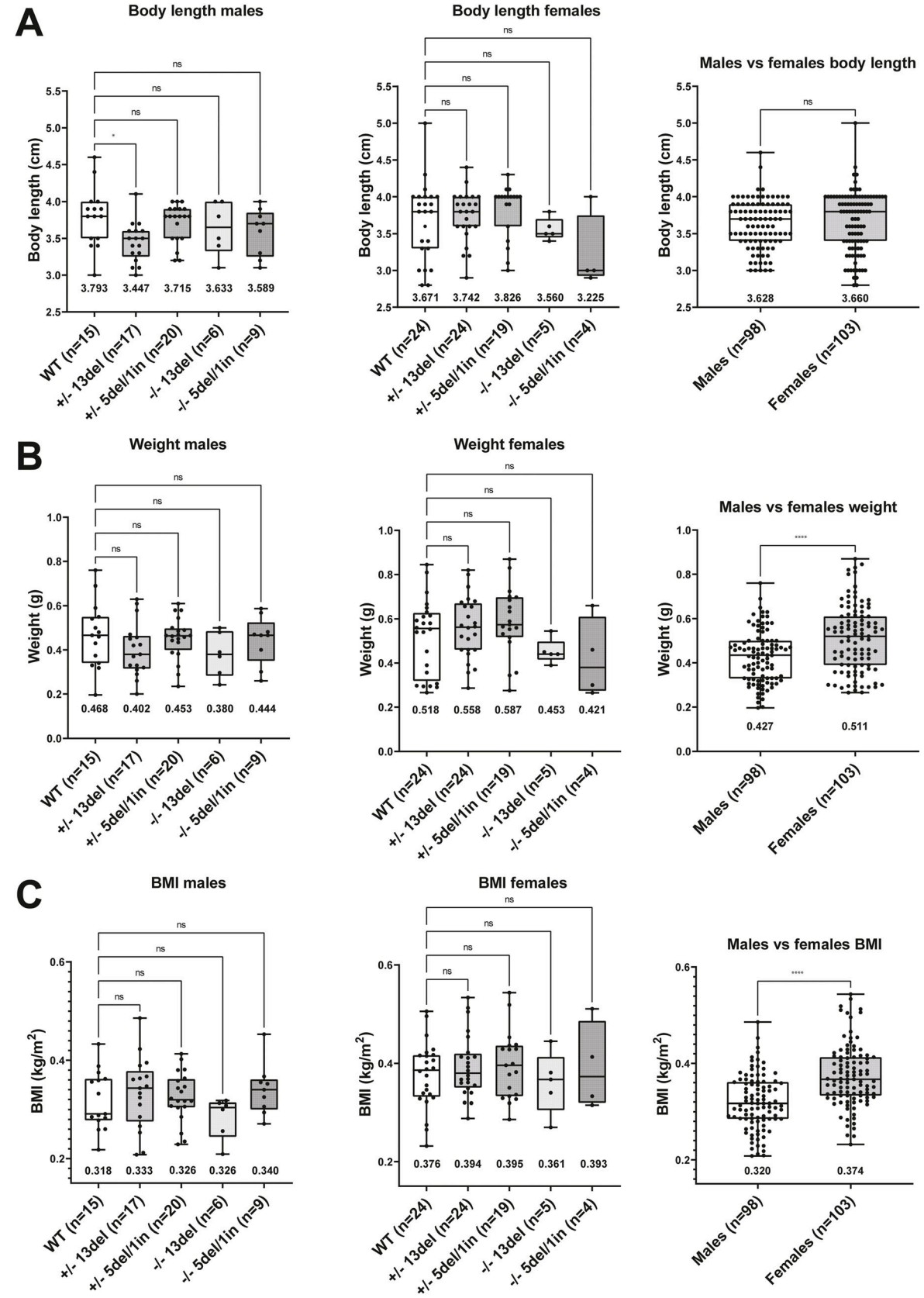

et al, 2022; Varland et al, 2023). The majority of Nt-acetylation events in eukaryotes is catalyzed by a set of five conserved NATs, NatA-NatE, acting co-translationally (Ree et al, 2018; Aksnes et al, 2023). Furthermore, posttranslational NAT enzymes were recently uncovered both in the plant and animal kingdoms (Van Damme et al, 2011a; Dinh et al, 2015; Drazic et al, 2018; Bienvenut et al, 2020). The two currently known posttranslational animal kingdom NATs are NatF/NAA60, acetylating transmembrane proteins (Van Damme et al, 2011b; Aksnes et al, 2015), and NatH/NAA80, acetylating actins (Drazic et al, 2018; Goris et al, 2018; Wiame et al, 2018). The NAT machinery and the Nt-acetylome have not been thoroughly studied in fish. Only basic studies on NatA and NatC have been conducted (Wenzlau et al, 2006; Ree et al, 2015). In this study, we explored the mechanism and impact of Nt-acetylation of actin, the most common cellular protein in animals with numerous functions. Our results demonstrate that, in zebrafish as in humans, Nt-acetylated actin is the predominant form in WT fish. Estimating the Nt-acetylation stoichiometry is not possible using our methodology, and targeted quantitation using isotopically labeled peptides would be required, as Nt-acetyl site occupancy is close to 100% in WT cells (Prus et al, 2019; Drazic et al, 2022). Furthermore, since we used a LysC/trypsin mix for the proteomics experiments, N-terminally arginylated peptides would not be detected if they existed (Drazic et al, 2022), as trypsin would remove the Nt-arginine. Actin Nt-acetylated peptides rank predominantly in the top $30^{th}$ percentile of peptides by intensity in *naa80+/+* and *naa80+/−* fish, with some in the $98^{th}$ percentile. Non-Nt-acetylated actin N-termini are absent in skeletal and cardiac muscle from *naa80+/+* and *naa80+/−* fish, suggesting a high Nt-acetylation stoichiometry. The absence of Nt-acetylated actin in all but one of the *naa80−/−* samples indicates that *naa80* status is the major determinant of actin Nt-acetylation in zebrafish heart and skeletal muscle. The low-intensity measurement of Nt-acetylated cytoplasmic actin 2 could stem from carryover; it might also be Nt-acetylated by Naa10 in *naa80−/−* fish, as zebrafish Naa10, like human NAA10, can Nt-acetylate acidic N-termini without an initiator methionine (Van Damme et al, 2011a; Ree et al, 2015). Alpha-cardiac muscle 1b actin was present with high relative intensity in the skeletal muscle samples, though cardiac actins are thought to be mainly expressed in heart muscle. If zebrafish alpha-cardiac 1b actin (N-terminus DDEETT) is classified as cardiac actin solely due to similarity to human cardiac actin (which also has the N-terminal sequence DDEETT [Fig 1]), and not due to expression data, it may be the case that this isoform is indeed expressed more highly in skeletal muscle. DDDETT (alpha-skeletal 1a) is slightly higher in heart muscle and could also be misclassified. More focused studies are needed to clarify this, as little research has been performed on actin isoforms in zebrafish.

This study represents the first animal knockout model of this recently characterized actin-modifying enzyme. Before phenotyping, several hypotheses emerged. One suggested that actin Nt-

acetylation would be critical to the blastula-gastrula stage cell morphogenetic movements, predicting that *naa80−/−* fish would not survive this stage, or alternatively, would exhibit significant morphological abnormalities. This hypothesis was not supported by our observations, indicating that Naa80 is not required for normal early development. A related hypothesis posited that actin Nt-acetylation might be crucial for normal actin dynamics during the blastula stage, and *naa80−/−* embryos from a *naa80+/−* mother may still have maternal *naa80* mRNA, allowing actin Nt-acetylation before the mid-blastula transition from maternal to zygotic gene expression. Maternally encoded Naa80 and Nt-acetylated actin would then be diluted in the developing embryo and eventually degraded. Post mid-blastula transition, Nt-acetylated actin would not be functionally present; if actin Nt-acetylation was only required during early morphogenetic movements this could still result in a viable embryo. However, embryos from *naa80* 13del −/− incrosses, which were viable and with no discernible morphological phenotype, suggest that maternal actin Nt-acetylation is not essential for normal development. Another hypothesis was that the loss of actin acetylation would alter actin-containing structures and organs. Cultured human cells lacking of *NAA80* and actin Nt-acetylation show increased motility and cell size, with unacetylated actin displaying altered polymerization properties in vitro (Drazic et al, 2018). Muscle cells depend on actin-myosin networks for their contractile function, but there have been no studies of muscle cells lacking actin acetylation. Our results indicate that under normal conditions there is no significant loss of function in muscle cells, as swimming and feeding behavior are comparable to that of heterozygous or WT siblings. Therefore, Nt-acetylation might not be necessary for functional actin networks under normal conditions in vivo.

Similarly, while we hypothesized that cardiomyocytes might be impaired by the loss of actin acetylation, we have no evidence suggesting diminished heart function in *naa80 −/−* zebrafish. One limitation of our study is the lack of stress testing such as swim-mill experiment, which might reveal decreased endurance in fish lacking Nt-acetylated actin. Individuals with pathogenic variants in actin genes exhibit various clinical phenotypes, including micro-cephaly, facial dysmorphism, intellectual disability, myopathy, and impaired hearing (Zhu et al, 2003; Kaindl et al, 2004; Rivière et al, 2012). Recently, two brothers with the same pathogenic *NAA80* variant were identified (Muffels et al, 2021). They exhibited phenotypes overlapping with those seen in individuals with impaired actin, such as craniofacial dysmorphisms, developmental delay, mild muscle weakness, and progressive high-frequency sensorineural hearing loss. This supports the primary role of NAA80 in actin Nt-acetylation in vivo and suggests that phenotypes driven by pathogenic *NAA80* variants may result from impaired actin function. Our *naa80 −/−* zebrafish showed reduced response to sound

**Figure 4. Body length and weight are not affected by *naa80* genotype.**
Adult zebrafish with the indicated genotype were weighed and measured from rostrum to caudal tip. A one-way ANOVA test followed up with a Dunnett's test was performed for the genotype comparison segregated by sex, where the +/+ values was selected as the control mean. *: $P < 0.05$, nonsignificant (ns): $P > 0.05$. Unpaired $t$ test was performed for male versus female comparison. ****: $P < 0.0001$, ns: $P > 0.05$. The mean value for each dataset is indicated beneath the boxplot. **(A)** Body length values for males and females. **(B)** Measured weight of males and females with indicated sample size and mean. **(C)** The BMI for males and females was calculated by using the formula BMI $= \frac{weight\,(kg)}{(body\,length\,(m))^{2}}$.

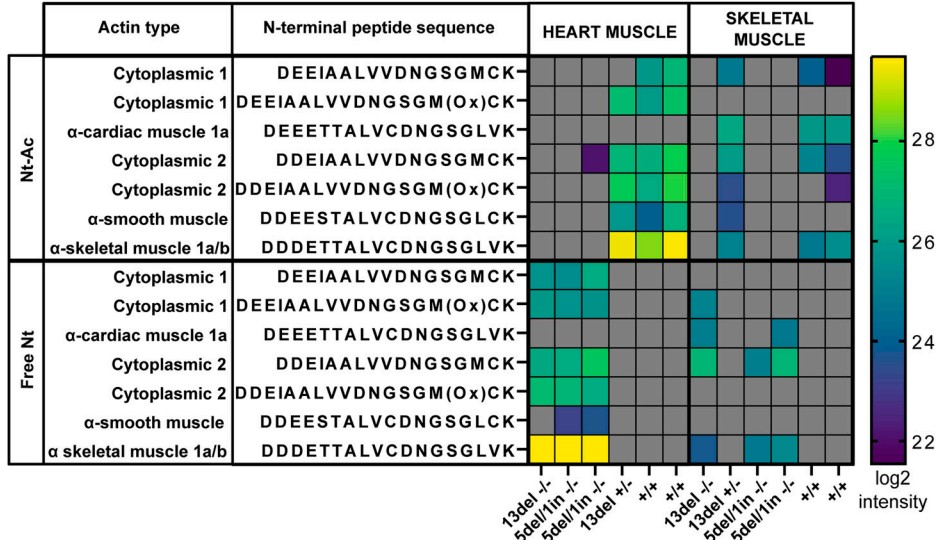

**Figure 5.   Defective in vivo actin Nt-acetylation in *naa80−/−* zebrafish.**
Mass spectrometry measurements of actin N-terminal peptides in cardiac muscle or skeletal muscle from the indicated genotypes. Log$_2$ intensity of the peptides is reported. Gray: not identified in the sample.

stimuli, and actin-related hearing phenotypes in humans have been particularly linked to defective γ-actin (Zhu et al, 2003). Thus, while many actin-related muscle functions may be normal in unstressed *naa80* −/− animals, Nt-acetylation could be crucial in other contexts, such as non-muscle actin dynamics in the ear.

No evidence has been published that Nt-acetylation affects the half-life of actin monomers. One study in human WT and *NAA80* knockout cells also found that actin is not significantly modified by N-terminal arginylation and is not likely to regulate actin function and thereby the hearing loss phenotype (Drazic et al, 2022). However, such *NAA80* knockout cells have been shown to have altered polymerization dynamics, altered cell morphology and increased speed of cell motility (Drazic et al, 2018), as well as Golgi fragmentation (Beigl et al, 2020). Specifically, unacetylated actin tends to both polymerize and depolymerize more slowly. These effects could all contribute to cell dysfunction and the specific phenotypes observed here. Focused studies of progenitor hair cells in the absence of Naa80 could potentially elucidate the mechanism.

# Materials and Methods

### Ethics statement

All experiments were approved by the Norwegian Food Safety Authority (application number 20/49856, approved 15.5.2020, with changes approved 17.2.2021 and 30.6.2022) and as per protocol 20-07 approved by the Institutional Animal Care Committee (IACUC) of Oklahoma Medical Research Foundation, Oklahoma City, USA.

### Zebrafish maintenance

Zebrafish were kept in the zebrafish facility at the Department of Bioscience, University of Bergen, Bergen, Norway or AALAC approved facility at Oklahoma Medical Research Foundation,

Oklahoma City, USA. Fish were kept at 28.5°C and a 14/10 h light/dark cycle. They were fed with artemia and powdered feed twice a day.

### Generation of two *naa80* knockout zebrafish lines: fin clipping, gDNA isolation and genotyping

Generation of zebrafish carrying *naa80* knockout alleles was performed by the Zebrafish Genetics Core Facility, The Hospital for Sick Children, Toronto, Canada. The gRNAs for *nat6/naa80* were designed using CHOPCHOP (Montague et al, 2014; Labun et al, 2016, 2019) following a previously described protocol (Varshney et al, 2016), and synthesized using the HiScribe T7 High Yield RNA Synthesis Kit (E2040S; NEB), following the manufacturer's instructions. The plasmid pT3TS-nCas9n (#46757; Addgene) was used as a template to synthesize Cas9 mRNA in vitro using the mMESSAGE mMACHINE T3 Transcription Kit (AM1348; Invitrogen). To generate F0 mosaic mutant animals, 100 pg of a *naa80*-targeting gRNA (CCAATGGCAGCGCAGCATGGGGG) was injected with 150 pg Cas9 mRNA into one-cell stage AB WT embryos at the concentration indicated above. Adult founders carrying a 5 bp deletion/1 bp insertion (5del/1in) or 13 bp deletion (13del) were identified by Sanger sequencing using the primers ACCGTCAAAGCACATAAGAACCT and GTAGTTAGGCACAATCGGGTACA. In the CDS of the now obsolete RefSeq record XM_005167127.3 (https://www.ncbi.nlm.nih.gov/nuccore/XM_005167127.3?report=genbank), the indel positions are: 107-111 GCAGC>A for the 5del/1in line and deletion of 101-113 GGCAGCGCAGCAT for the 13del line. Each line was incrossed to obtain F1 embryos which were shipped to the zebrafish core facility in Bergen. Adult zebrafish were anesthetized by placing in 40 mg/l buffered tricaine in system water for 2–4 min until gill movement was reduced, placed on a petri dish, weighed, and measured. The distal part of the tail fin was cut off with a clean scalpel, and the fish was immediately placed in system water to recover. The fish were either placed alone, with a mate of the opposite sex, or with two Spotty wild-type fish, while the genotyping was performed. gDNA

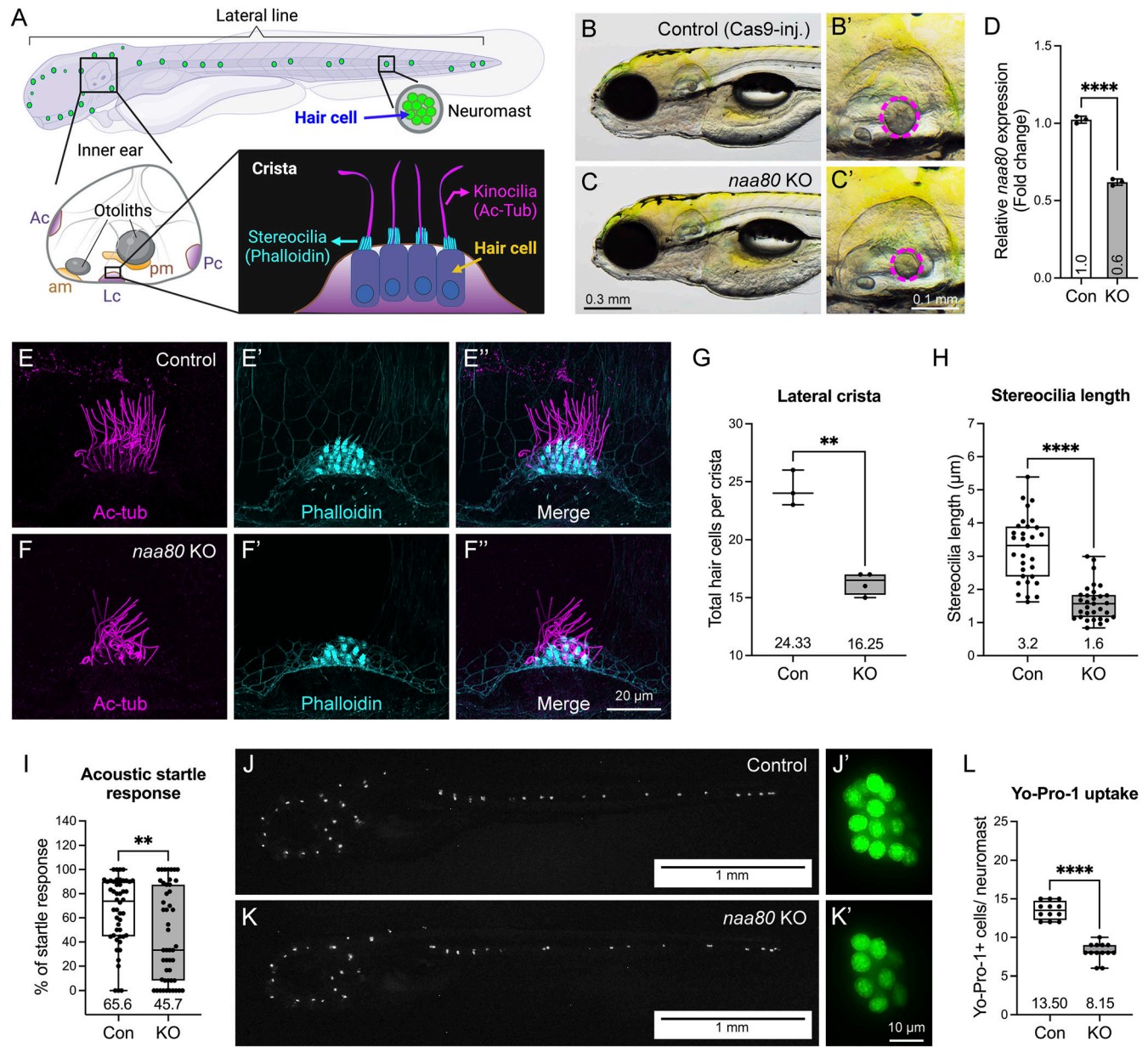

**Figure 6. Zebrafish *naa80* F0 KO larvae showed hearing loss-related phenotypes.**
**(A)** Schematic representation depicts sensory tissues in zebrafish larva at 5–6 dpf including inner ear and neuromasts. Five sensory patches of inner ear including anterior crista (Ac), lateral crista (Lc), posterior crista (Pc), anterior macula (am), and posterior macula (pm). Schematic representation of lateral crista at lower right corner. Kinocilia and stereocilia of hair cell can be revealed by immunohistochemistry using anti-acetylated tubulin Ac-tub and phalloidin (F-actin), respectively. **(B, C)** *naa80* F0 KO larvae have no obvious morphological abnormalities but showed smaller otoliths compared with Cas9-injected control larvae. **(D)** RT-qPCR showed a down-regulated *naa80* expression in F0 KO (n = 3). Expression levels were normalized to *18S* housekeeping gene and compared with the Cas9-injected controls. **(E, F, E', F', E", F")** Immunohistochemistry of both anti-acetylated tubulin (magenta, (E, F)) and phalloidin (cyan, E', F'), and merged (E", F"). **(G)** Quantification of hair cell numbers in the lateral crista of control (n = 3 larvae) and KO (n = 4 larvae) larvae. **(H)** Stereocilia length measurement in lateral crista hair cells of control and KO larvae. n = 31 stereocilia from three larvae for each group. **(I)** Evaluation of acoustic startle response. n = 48 larvae of each group. **(J, K)** Representative fluorescent images of Cas9-injected control (n = 12 larvae) and *naa80* F0 KO (n = 13 larvae) larvae after Yo-Pro-1 uptake. Left panel (J, K) was whole-mount images after invert color into black and white, anterior to the left and dorsal to the top. Right panel (J' and K') was enlarged single neuromast. **(L)** Quantification of Yo-Pro-1 positive cells per neuromast. Two-tailed unpaired *t* test with Welch's correction. **P < 0.01 and ****P < 0.0001.

was extracted from the fin essentially as described by (Samarut et al, 2016). The fin piece was boiled for 10 min in 18 $\mu$l 50 mM NaOH. 2 $\mu$l 100 mM Tris–HCl, pH 8.0 was added and the sample was centrifuged for 13,000*g* for 5 min. The supernatant was diluted either 1:10 or to 50 ng/$\mu$l in 100 mM Tris–HCl, pH 7.4. For sequencing, 1 $\mu$l was used for a PCR reaction with primers GTAGTTAGGCA-CAATCGGGTACA and GTCAGCTGCACAGTCTTTCG producing a 359 bp fragment containing the mutation site. The PCR reaction ran for 35

cycles with a Tm of 58.4°C. The PCR was checked by running on a 2% agarose/TAE gel with Gelred, and 1 μl of the PCR product was used in a Big Dye v3.1 (Applied Biosystems) sequencing reaction. Alternatively, genotyping for the 13del line was performed by agarose gel electrophoresis. 1 μl 1:10 diluted gDNA was used in a PCR reaction with primers GTAGTTAGGCACAATCGGGTACA and GGCGCTGTCTGATCTACTCC yielding a 112 bp product and a 62°C Tm. PCR products were run on a 4% agarose gel with 10 mM lithium borate acetate buffer and ethidium bromide, allowing the resolution of the 13 bp difference between the mutant and WT alleles. Coding sequences of the WT, 5 del/1 in and 13 del alleles, as well as the predicted protein sequences, can be found in Table S4.

### Actin Nt-acetylation status determination by mass spectrometry

Genotyped *naa80* +/+, +/− and −/− adult fish (F1) were euthanized and dissected, collecting lateral muscle, heart, and gut tissues. Fish were kept on ice after euthanasia and dissected tissue was flash-frozen in liquid nitrogen and kept at –80°C until processing. Frozen tissues were transferred to a Petri dish on ice and chopped finely with a clean scalpel. The tissue pieces were transferred to a preweighed microcentrifuge tube and lysed in 10 μl lysis buffer per mg tissue. Lysis buffer was composed of 50 mM Tris–HCl, pH 7.4, 150 mM NaCl and 1% NP-40, supplemented with 1 tablet/50 ml of cOmplete, EDTA-free Protease Inhibitor Cocktail (Roche) and 250 U/ml benzonase. Lysis was performed for 1 h on a rotating wheel at 4°C. The lysate was cleared by centrifugation at 17,000$g$ for 5 min, and the concentration was measured in the supernatant using the Pierce BCA kit according to manufacturers' protocol. To reduce and alkylate the proteins, 50 μg of protein sample was combined with an equal volume of alkylation buffer containing 4% SDS, 10 mM TCEP, 20 mM chloroacetamide, and 200 mM Tris–HCl, pH 8.0, and heated to 95°C for 10 min. To digest and clean up proteins in the supernatant, we performed protein aggregation and capture (PAC) essentially as described (Batth et al, 2019). 250 μg Sera-Mag carboxylate-modified hydrophilic and hydrophobic magnetic beads (Cytiva) in a 50:50 ratio in acetonitrile were added to 50 μg alkylated protein, so that the final concentration of acetonitrile was 70% and the protein:bead ratio was 5:1. The samples were vortexed, left to settle for 10 min, vortexed, and settled for 10 more minutes, and then the beads with aggregated proteins were separated from the supernatant with a magnet. The supernatant was removed with vacuum suction, and 1 ml 100% acetonitrile was added while the tubes were still on the magnet, care being taken not to disturb the beads. The acetonitrile was removed and the beads were washed on the magnet twice with 1 ml 70% ethanol. The beads were then resuspended in 250 μl digestion solution containing 1 μg trypsin and 0.5 μg LysC in 25 mM Tris–HCl, pH 8.0. The beads were incubated for 16 h at 37°C and shaking at 1,100 rpm. 1/10 volume of 10% TFA was added to each sample to acidify it and stop the digestion. The beads were separated on the magnet and the supernatant was transferred to a fresh tube, centrifuged at 13,000$g$ for 1 min and transferred to a new tube to remove any residual beads, and desalted using a 1 ml C18 Sep-pak (Waters) according to the manufacturer's instructions.

Peptides were dried in a speedvac, resuspended in 0.1% formic acid, and stored at –20°C until analysis.

### LC/MS for zebrafish tissue proteomics

About 0.5 μg protein as tryptic peptides dissolved in 2% acetonitrile (ACN), 0.5% formic acid (FA), were injected into an Ultimate 3000 RSLC system (Thermo Fisher Scientific) connected online to a Exploris 480 mass spectrometer (Thermo Fisher Scientific) equipped with EASY-spray nano-electrospray ion source (Thermo Fisher Scientific). The sample was loaded and desalted on a pre-column (Acclaim PepMap 100, 2 cm × 75 μm ID nanoViper column, packed with 3 μm C18 beads) at a flow rate of 5 μl/min for 5 min with 0.1% trifluoroacetic acid. Peptides were separated during a biphasic ACN gradient from two nanoflow UPLC pumps (flow rate of 200 nl/min) on a 50 cm analytical column (PepMap RSLC, 50 cm × 75 μm ID EASY-spray column, packed with 2 μm C18 beads). Solvent A and B were 0.1% TFA (vol/vol) in water and 100% ACN, respectively. The gradient composition was 5% B during trapping (5 min) followed by 5–8% B over 1 min, 8–25% B for the next 124 min, 25–36% B over 30 min, and 36–80% B over 5 min. Elution of very hydrophobic peptides and conditioning of the column were performed during 10 min isocratic elution with 80% B and 15 min isocratic conditioning with 5% B. Instrument control was through Thermo Fisher Scientific SII for Xcalibur 1.6. The eluting peptides from the LC-column were ionized in the electrospray and analyzed by the Orbitrap Exploris 480. The mass spectrometer was operated in the DDA-mode (data-dependent-acquisition) to automatically switch between full scan MS and MS/MS acquisition. Instrument control was through Orbitrap Exploris 480 Tune 3.1 and Xcalibur 4.4. MS spectra were acquired in the scan range 350–1,400 m/z with RF lens at 40%, resolution R = 120,000 at m/z 200, automatic gain control (AGC) target of 3 × 10$^6$ and a maximum injection time (IT) at "Auto." The 15 most intense eluting peptides above an intensity threshold of 50,000 counts, and charge states 2–5 were sequentially isolated to a target value (AGC) of 1 × 10$^5$ and a maximum IT of 75 ms in the C-trap, and isolation width maintained at 1.2 m/z, before fragmentation in the higher-energy collision dissociation (HCD) cell. Fragmentation was performed with a normalized collision energy (NCE) of 30%, and fragments were detected in the Orbitrap at a resolution of 30,000 at m/z 200, with first mass fixed at m/z 110. One MS/MS spectrum of a precursor mass was allowed before dynamic exclusion for 45 s with "exclude isotopes" on. Lock-mass internal calibration was not used. The spray and ion-source parameters were as follows. Ion spray voltage = 1,800 V, no sheath and auxiliary gas flow, and capillary temperature = 275°C.

### Database searching of proteomics data

Data were processed in Fragpipe (v. 17.1) using the LFQ-MBR workflow. The enzyme specificity was set to semi-specific trypsin (free N-terminus). The data were searched against a zebrafish proteome database containing 20,366 sequences (3,235 annotated in Swiss-Prot and 17,131 unreviewed TrEMBL sequences). Carbamidomethylation of cysteine was set as a fixed modification and N-terminal acetylation of peptide and protein N-termini as well as methionine oxidation were set as variable modifications. The

combined_modified_peptide.tsv output file was used for further analysis to determine the acetylation status of N-terminal actin actin peptides.

### Generation of zebrafish founder knockout (KO) larvae for phenotyping

Two single guide RNA (sgRNA) target sequences (5′-ACTCAA-CATGAAAAGGTCAT-3′ with CGG PAM and 5′-TATGGCCGAATCCT-TATGGA-3′ with AGG PAM) were designed using the CRISPOR tool (González et al, 2022) and chemically synthesized by Synthego Inc. The sgRNAs, along with 1 $\mu l$ of 40 $\mu M$ Cas9-NLS protein from UC Berkeley QB3 Macrolab, were combined with 500 ng of each sgRNA (in 3 $\mu l$) and 2 $\mu l$ of 1 M potassium chloride. This mixture was injected into one-cell stage WT embryos. Phenotypic analysis was conducted on the founder generation of knockouts at the 5-day post-fertilization (dpf) stage.

### WISH

WISH was conducted utilizing *naa80* antisense probes labeled with digoxigenin, which were synthesized from a PCR-amplified template derived from zebrafish cDNA containing T7 promoter sequences at the 3′ end (Thisse & Thisse, 2008). The specific primer sequences used were Forward 5′- GTACCCGATTGTGCCTAACT-3′ and Reverse 5′- GAATTGTAATACGACTCACTATAGGGCCGAGTCTCA-GATGTCCTTG-3′. RNA synthesis was performed using T7 RNA polymerase from Roche.

### RNA extraction and reverse transcription-quantitative PCR (RT-qPCR)

RNA was extracted from various developmental stages and adult tissues using TRIzol Reagent (Thermo Fisher Scientific) and further purified with the RNA clean and concentrator-5 kit (Zymo), following the provided instructions. Subsequently, RNA samples were reverse transcribed into cDNA using the iScript cDNA-synthesis kit (Bio-Rad), as per the manufacturer's protocol. The generated cDNA served as a template for RT-qPCR reactions, performed with SYBR Green Supermix (Thermo Fisher Scientific) and the Light Cycler 96 System (Roche) according to the manufacturer's guidelines. Each amplification was conducted with three technical replicates and normalized to the *18S* housekeeping gene. The primer sequences used were Forward (5′- CTATCCCGTGTGTCTCCTGC-3′) and Reverse (5′- TGCAAACCACTACCGACTCC-3′). Cycle threshold values (Ct) data were analyzed for relative gene expression using Microsoft Excel. Quantification was performed using the $2^{(-\Delta\Delta CT)}$ method, with 18 hpf and liver samples used as calibrators for different developmental stages and adult tissues, respectively.

### Acoustic startle response assay

The Acoustic startle response test was conducted in Zebrabox behavior chambers (Viewpoint Life Sciences) at the standard RT, following the previously described protocol (Lin et al, 2021). The test assessed the percentage of responses to 12 stimuli per larva.

### Immunohistochemistry

The hair cell stereocilia in the inner ear were labeled for F-actin using fluorescently tagged (FITC) phalloidin dye, and the kinocilia were stained for anti-acetylated tubulin at 5 dpf, following the method previously outlined (Lin et al, 2021). Subsequently, the stained larvae were immersed in 75% glycerol, and images were captured using a Zeiss LSM-710 confocal microscope.

### Yo-Pro-1 uptake

Whole-mount staining of live zebrafish larvae was performed using Yo-Pro-1 (Invitrogen) to observe neuromast hair cells. Zebrafish larvae at 5 dpf were anesthetized with tricaine and treated with 1 mM Yo-Pro-1 diluted in 5 ml of embryo medium for 1 h at RT to allow thorough probe penetration.

### Expression of zebrafish Naa80-MPB

To clone zebrafish *naa80*, a cDNA library was constructed by isolating total mRNA from a pool of 5 dpf TAB WT embryos essentially as described (Ree et al, 2015). The embryos were lysed in Trizol and RNA was extracted using chloroform/phenol followed by isopropanol precipitation. The library was generated using the Transcriptor Reverse Transcriptase kit (Roche) and poly-dT and random hexamer primers. *naa80* was amplified by PCR and subcloned into the pETM41 vector, which encodes an N-terminal His-tag and MBP tag, ampicillin resistance, and is inducible by IPTG. pETM41-*naa80* was introduced into *E. coli* BL21* cells (Invitrogen) by heatshock transformation. Heatshocked cells were grown in rich medium for 1 h at 250 rpm before plating on lysogeny broth-ampicillin (LB-amp) plates and overnight growth. From this, we made a preculture and from this a glycerol stock, which was kept at −80°C. The glycerol stock was sequenced to verify the plasmid sequence. To express MBP-Naa80 for purification, a preculture of 5 ml was inoculated with the glycerol stock, and the cells were grown at 250 rpm and 37°C for 16 h, then diluted in a 500 ml culture and grown until they reached an optical density of 0.6 at 600 nm. Protein expression was induced by addition of 1 mM IPTG. The cells were further cultured at 20°C and 250 rpm overnight, then harvested by centrifugation at 3,095$g$ for 30 min. The cell pellets were either stored at −20°C or lysed immediately. Lysis was performed by dissolving in 10 ml ice-cold lysis buffer (20 mM imidazole, 50 mM Tris–HCl, pH 8.0, 300 mM NaCl, 1x complete EDTA-free protease inhibitor cocktail) and sonicating on ice for 7 min with amplitude 55 and 1 s pulses. The cell lysate was then centrifuged for 15 min at 17,500$g$. The supernatant was loaded on a HisTrap 5 ml nickel-NTA column (GE Healthcare) with a peristaltic pump. The HisTrap was transferred to an ÄKTA Pure system and column was washed with 50 ml wash buffer (same as lysis buffer with no protease inhibitor) and eluted with a high imidazole buffer (300 mM imidazole, 50 mM Tris–HCl, pH 8.0, 300 mM NaCl) into a 96-deepwell plate. The relevant fractions were dialyzed into gel filtration buffer (50 mM Tris–HCl, pH 8.0, 300 mM NaCl, and 1 mM DTT. Filtered and degassed) overnight with a 12 kD dialysis membrane, and then concentrated to a volume of 2 ml. Gel filtration chromatography was performed on the ÄKTA Pure with a Superdex 200 16/600 column and gel filtration

buffer. The relevant fractions were concentrated, concentration was measured using the BCA Protein Assay Kit (Pierce) and MBP-Naa80 was diluted to 2 $\mu$M in gel filtration buffer. For long-term storage, protein was frozen in either 50% glycerol and stored at –20°C or 10% glycerol and stored in aliquots at –80°C. Purity was checked by SDS–PAGE.

### N-terminal acetylation assays

N-terminal acetylation activity was measured using carbon-14 labeled Ac-CoA as described (Drazic & Arnesen, 2017). Briefly, 250 nM purified MBP-Naa80, 200 $\mu$M substrate peptide (Fig 1 and Table S5, all supplied by Innovagene at >95% purity), 50 $\mu$M $^{14}$C-Ac-CoA (Perkin Elmer) were mixed in acetylation buffer (50 mM Tris–HCl pH 8.5, 10% glycerol, 1 mM DTT and 0.2 mM EDTA) in a 25 $\mu$l volume. Reaction proceeded for 1 h at 37°C. 20 $\mu$l per sample was transferred to a 1 cm$^2$ piece of P81 filter paper. All pieces were washed 3 × 5 min in 10 mM Hepes, pH 7.4 to remove unincorporated $^{14}$C-Ac-CoA, and air dried. The dry filter papers were placed in scintillation vials with 5 ml Ultima Gold F scintillation cocktail (Perkin Elmer) and scintillation was measured in a TriCarb 2900TR liquid scintillation analyzer (Perkin Elmer).

## Data Availability

All MS data are deposited to the ProteomeXchange Consortium, via the PRIDE (https://www.ebi.ac.uk/pride/) partner repository with the dataset identifier PXD046978.

## Supplementary Information

## Acknowledgements

Mass spectrometry-based proteomic analyses were performed by the Proteomics Unit at the University of Bergen (PROBE). This facility is a member of the National Network of Advanced Proteomics Infrastructure (NAPI), which is funded by the Research Council of Norway (INFRASTRUKTUR-program project number: 295910). This work was supported by grants from the Norwegian Health Authorities of Western Norway (F-12540 to T Arnesen), and the European Research Council (ERC) under the European Union Horizon 2020 Research and Innovation Program (Grant 772039 to T Arnesen), and funds from the Oklahoma Medical Research Foundation, Oklahoma City, OK, USA (GK Varshney)

### Author Contributions

R Ree: conceptualization, supervision, investigation, visualization, methodology, and writing—original draft, review, and editing.
S-J Lin: investigation, visualization, methodology, and writing—original draft, review, and editing.
LO Sti Dahl: investigation.
K Huang: investigation.

C Petree: investigation.
GK Varshney: conceptualization, supervision, funding acquisition, investigation, methodology, project administration, and writing—original draft, review, and editing.
T Arnesen: conceptualization, supervision, funding acquisition, project administration, and writing—original draft, review, and editing.

### Conflict of Interest Statement

The authors declare that they have no conflict of interest.

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
