## [Reviewer comments · Life Science Alliance]

Life Science Alliance

Naa80 is required for actin N-terminal acetylation and normal hearing in zebrafish

Rasmus Ree, Sheng-Jia Lin, Lars Ole Sti Dahl, Kevin Huang, Cassidy Petree, Gaurav Varshney, and Thomas Arnesen
DOI: <https://doi.org/10.26508/lsa.202402795>

Corresponding author(s): Rasmus Ree, NORCE Norwegian Research Centre and Thomas Arnesen, Department of Biomedicine, University of Bergen, Bergen, Norway

Review Timeline:

Submission Date:	2024-04-25
Editorial Decision:	2024-05-24
Revision Received:	2024-09-19
Editorial Decision:	2024-09-20
Revision Received:	2024-09-27
Accepted:	2024-09-27

Transaction Report:

May 24, 2024

Re: Life Science Alliance manuscript #LSA-2024-02795-T

Dr. Rasmus Ree
NORCE Norwegian Research Centre AS
Norway

Dear Dr. Ree,

Thank you for submitting your manuscript entitled "Naa80 is required for actin N-terminal acetylation and normal hearing in zebrafish" to Life Science Alliance. The manuscript was assessed by expert reviewers, whose comments are appended to this letter. We invite you to submit a revised manuscript addressing the Reviewer comments.

Thank you for this interesting contribution to Life Science Alliance. We are looking forward to receiving your revised manuscript.

Sincerely,

B. MANUSCRIPT ORGANIZATION AND FORMATTING:

Reviewer #1 (Comments to the Authors (Required)):

Maturation of actin requires N-terminal acetylation, and NAA80 was recently identified as an enzyme responsible for this specific post-translational modification of actin. NAA80 knockout cells display various defects, but the in vivo roles of NAA80-catalyzed N-terminal acetylation of actin have not been reported so far. Here, Ree et al. demonstrate that the zebrafish orthologue of NAA80 (zNaa80) acetylates actin. By analyzing zNaa80 knockout zebrafish, they provide evidence that zNaa80 is critical for actin N-terminal acetylation in vivo. Interestingly, their results provide evidence that the N-terminal acetylation of actin is not essential for the normal development or morphology of zebrafish. However, loss of zNaa80 results in defects in otolith size and stereocilia bundles, and in accompanying hearing defects.

The results of this study are interesting and provide important new information on the in vivo roles of NAA80 and N-terminal acetylation of actin. However, the presentation of the data is in many cases somewhat confusing, and for example several important control experiments are not shown in the manuscript. Thus, extensive revision of the manuscript text and figures is required before the manuscript is acceptable for publication.

Specific comments:

1. The authors have used two zNaa80 knockout lines in this study, but how these knockouts were confirmed and validated was not adequately described in the manuscript. The authors just state in the text that 'The knockout alleles were verified by PCR using naa80-specific primers (see Table 1) followed by either Big Dye sequencing or agarose gel electrophoresis', but none of these data were presented in the manuscript. Thus, the authors should prepare a new figure, where validation of the knockout lines would be thoroughly presented. Here, as well as in the 'Results' section, the authors should also clearly describe mutations in the 'naa80 13del' and 'naa80 5del/1in' lines, because in the current version of the text this is quite confusing.
2. The rationale of using transient CRISPR method to study the role of zNaa80 in hearing functions is unclear from the current version of the text. The authors should thus clearly explain why the stable mutant lines were suitable for the analyses presented in Figs. 3 and 4, but not for the subsequent studies focusing on otolith morphology and hearing problems. Additionally, the success of the transient CRISPR method should be validated, and the loss of functional zNaa80 (at the genomic, mRNA, or protein level) should be presented as an additional figure or supplementary figure.
3. The authors state in the Discussion that 'under normal conditions there is no major loss of function, as swimming and feeding behavior is not appreciably different from heterozygous or wildtype tankmates'. Again, these data were not presented in the manuscript, and should be shown.
4. The differences in the N-terminal acetylation of actin in the heart and skeletal muscle tissues of wild-type and zNaa80 mutants are presented in Fig. 4. If technically feasible, it would be important to carry out similar analysis also on some non-muscle tissues (e.g. brains) to confirm that also in these tissues N-terminal acetylation of actin requires zNaa80.
5. The figure legends lack essential information in certain cases. For example, the legend to Fig. 5 G-H reads 'Representative image of Yo-Pro-1 uptake.', but what is presented in the four panels remains elusive.
6. Fig. 1C: The text on the y-axis of the graph is 'Activity normalized to DDEI', but from the figure the activity appears to be normalized to DEEI. This should be corrected/clarified.
7. Manuscript text contains many typos and awkward sentences, and thus the text should be carefully edited.

Reviewer #2 (Comments to the Authors (Required)):

Summary

NAA80 is an acetyltransferase that is responsible for unique N-terminal acetylation of actin. This study reports characterization of phenotypes of NAA80 mutants in zebrafish, which is the first animal model of this conserved enzyme. They demonstrated by

mass spec that N-terminal acetylation of both muscle and cytoplasmic actin isoforms were absent, indicating that NAA80 is required for this actin modification *in vivo*. Surprisingly, the mutant fish were superficially normal in morphology, although they reported poor breeding of unknown mechanism. They found that the NAA80-null fish have fewer actin-based stereocilia in the inner ear than wild-type and exhibit impaired acoustic startle response. This phenotype might be relevant to the hearing loss that is linked to NAA80 missense mutations in humans. Overall, this work should be an important contribution to the cell biology community because the zebrafish model of NAA80 mutation could be a valuable resource to advance our knowledge on the functional significance of the unique actin acetylation. I have a few minor suggestions as described below.

1. In Fig. 5, reduced number of hair cell bundles in the NAA80 mutants is an important and interesting observation. Is this because the number of hair cells was reduced, or the number of stereocilia bundles per hair cell was reduced? When the number of stereocilia was reduced, how was the number/organization of kinocilia affected?
2. In Fig. 4, the results of mass spec should be strengthened if the authors can demonstrate that acetylation of other control proteins were not affected by the NAA80 mutation.

Reviewer #3 (Comments to the Authors (Required)):

In this manuscript, Ree et al. identified the *naa80* gene in zebrafish as an ortholog of the NAA80 gene in humans and evaluated the N-terminal acetylation activity of the product. Furthermore, the authors developed zebrafish strains lacking functional Naa80 proteins and described the phenotype. Although the authors' zebrafish model does not show significant morphological or behavioral phenotypes, the authors carefully investigate this zebrafish model and point out that the *naa80*-null zebrafish show morphological and/or functional abnormality in the inner ear and the lateral line. The phenotype in hair cells may be a clue to elucidate how variants of the NAA80 gene cause progressive high-frequency hearing loss. From these aspects, I agree that this study will be of interest to researchers studying actin dynamics and hearing and recommend this manuscript for publishing in the Life Science Alliance journal if the following issues are addressed properly.

1. The authors identify the *naa80* gene as a zebrafish ortholog of human NAA80 and demonstrate the N-terminal acetylation activity of the Naa80 protein. However, it is not clear how the amino-acid sequence of the zebrafish Naa80 protein is similar to the amino-acid sequence of NAA80 proteins of other species including humans and mice. I request adding a figure to Figure 1 (or preparing a supplemental figure) and comparing amino-acid sequences between species.
2. The authors develop two *naa80*-null zebrafish alleles, 13del or 5del/1in, and report that *naa80*-null zebrafish may have difficulty in laying eggs. However, it is not clear how 13del and 5del/1in alleles disables the function of the *naa80* gene. I request adding a scheme to Figure 3 (or preparing a supplemental figure) to describe the modification(s) that the authors made to this gene, especially on a map of exons and introns. In addition, it is difficult to track how the authors crossed the zebrafish with the 13del or 5del/1in alleles and found the phenotype in laying eggs. I request preparing a supplemental figure illustrating the breeding described in the first and second paragraphs of the "*naa80* -/- zebrafish are free from gross morphological phenotypes and develop normally" section.
3. The authors reported that *naa80*-null zebrafish display a reduced size of otolithic organ(s) in the inner ear, decreased number of bundles, weaker acoustic startle responses and reduced uptake of Yo-Pro-1. For these phenotypes, I request more details including (1) clarification of "bundles" in Figure 5E (are these kinocilia?) and (2) evaluation of stereocilia morphology (at least their length).
4. Please discuss the possible scenarios of how dysfunction of *naa80* causes phenotypes in hair cells and otoliths. Do you think degradation of actin monomers is related to this phenotype? Do you think the speeds of polymerization and depolymerization are associated with the phenotypes?

Minor points:

Page 5, Figure 5E and 5F:

The titles of these two graphs don't seem to represent the data in the graphs. Is it possible to state like, "*naa80* expression level during the development of zebrafish" and "*naa80* expression levels in adult zebrafish tissues"?

Page 6, "Male *naa80* 13del+/- fish were significantly shorter than controls ($p < 0.05$); however, *naa80* 13del-/- fish were not significantly shorter so the *naa80* genotype is not likely to be the cause of this.":

The term "short" may not be scientific. Is it possible to state like, "The body length of male *naa80* 13del+/- fish were (are)..."?

Page 8, the "Impaired N-terminal acetylation of cytoplasmic and muscle actins in *naa80* -/- zebrafish" section; Page 9, the "*naa80*-/- F0 knockout (KO) zebrafish displayed hearing-related defects" section; Page 10, Figure 5

In these sections and figures, it is uncertain which of the 13del-/- and 5del/1in-/- lines are used. Please clarify the lines used in these sections and figures. If both of these two lines are used, please separate the data.

Page 10, "However, we did observe a reduction in otolith size compared to controls following Cas9 protein injection (Figure 5B)."; Page 10 Figure 5 legend:

The phrases, "reduction in otolith size" and "smaller otoliths", sound like the size of otoliths are small. I suspect that the authors intend to clarify that "the area covered by otolith" is small. Please rephrase if my understanding is correct. This is confusing since giant otoliths are observed in some diseases.

We would like to thank the reviewers for providing helpful and insightful comments on our manuscript "*Naa80 is required for actin N-terminal acetylation and normal hearing in zebrafish*" (manuscript number LSA-2024-02795-T), which we submitted to Life Science Alliance in April.

Below are our responses (in blue) to the specific points raised by the reviewers. Find enclosed as well the revised manuscript, high-resolution figures and supplemental tables.

Reviewer #1

1. The authors have used two zNaa80 knockout lines in this study, but how these knockouts were confirmed and validated was not adequately described in the manuscript. The authors just state in the text that 'The knockout alleles were verified by PCR using naa80-specific primers (see Table 1) followed by either Big Dye sequencing or agarose gel electrophoresis', but none of these data were presented in the manuscript. Thus, the authors should prepare a new figure, where validation of the knockout lines would be thoroughly presented. Here, as well as in the 'Results' section, the authors should also clearly describe mutations in the 'naa80 13del' and 'naa80 5del/1in' lines, because in the current version of the text this is quite confusing.

The validation of the stable KO lines is now shown in new Figure 3C and Supplemental Figure S1. Figure 3C shows a representative agarose gel used to screen for fish carrying the 13del allele, while Figure S1 C shows representative DNA sequencing chromatograms used to genotype fish carrying either allele. Further, new Figure 3B and new Supplemental Figure S2 show the predicted gene products of each mutant allele, which are predicted to completely disrupt protein function. The coding sequences and predicted protein sequences are provided in new Supplemental Table S5.

2. The rationale of using transient CRISPR method to study the role of zNaa80 in hearing functions is unclear from the current version of the text. The authors should thus clearly explain why the stable mutant lines were suitable for the analyses presented in Figs. 3 and 4, but not for the subsequent studies focusing on otolith morphology and hearing problems. Additionally, the success of the transient CRISPR method should be validated, and the loss of functional zNaa80 (at the genomic, mRNA, or protein level) should be presented as an additional figure or supplementary figure.

Experiments with the stable lines were performed in Bergen, by the Arnesen group, and included proteomics and gross morphological phenotyping. Experiments on hearing problems and otolith morphology, using the transient CRISPR method, were performed in Oklahoma by the Varshney group. Ideally, we would also have been able to investigate the hearing phenotype in the stable lines, which could have strengthened the findings. However, the project was ending in Bergen and there were no resources to do this.

Regarding validation of the transient CRISPR line, the new Figure 6D shows lower levels of *naa80* mRNA as measured by qPCR in the *naa80* F0 KO larvae.

3. The authors state in the Discussion that 'under normal conditions there is no major loss of function, as swimming and feeding behavior is not appreciably different from heterozygous or wildtype tankmates'. Again, these data were not presented in the manuscript, and should be shown.

We had a working hypothesis that morphology, swimming or feeding behavior may be affected by dysregulated actin Nt-acetylation. While observing the fish in the tanks throughout their development, we did not see any such effects of the KO. Our conclusions regarding normal gross morphology, swimming function and eating behavior is thus based on observing no differences in morphology or these behaviors of the *naa80* KO fish. Since we did not observe any differences, we did not document this beyond body weight and length measurements. Body weight and length were not significantly different between the different genotypes in F2. This additionally supports the assertion about feeding behavior, as problems with taking in food would likely affect body weight, body composition and viability.

4. The differences in the N-terminal acetylation of actin in the heart and skeletal muscle tissues of wild-type and zNaa80 mutants are presented in Fig. 4. If technically feasible, it would be important to carry out similar analysis also on some non-muscle tissues (e.g. brains) to confirm that also in these tissues N-terminal acetylation of actin requires zNaa80.

We chose heart, skeletal muscle and gut tissue for the proteomics analyses to obtain data on as many actin isoforms as possible – non-muscle cytoplasmic actin, cardiac and skeletal muscle actins, and smooth muscle actins. For reasons which are not clear, the proteome coverage was low in the gut samples, and no actin N-termini were identified. Nevertheless, as shown in new Figure 5 (previous Figure 4), the Nt-acetylation of all the muscle and non-muscle actin N-termini we identified depended on having one functional *naa80* allele. This covered cytoplasmic type I actins, which are found in all cells, as well as cardiac, skeletal muscle and smooth muscle actins. Smooth muscle actins are likely present in vascular tissue in muscle tissues as well, potentially explaining their presence in heart and to a lesser extent skeletal muscle samples. This strongly suggests that Naa80 catalyzes this modification across tissues; the alternative hypothesis being there is additional enzymatic machinery performing it in other tissues, but none of the ones we sampled. We cannot rule this out although it is highly unlikely.

None of the authors have the capacity to carry out these additional experiments. We have added the qualifier that the Nt-acetylation of actin by Naa80 has only been shown in muscle and heart tissue. This is added to the introduction (line 68-69), the relevant results section (line 200) and the discussion (lines 261 and 264).

5. The figure legends lack essential information in certain cases. For example, the legend to Fig. 5 G-H reads 'Representative image of Yo-Pro-1 uptake.', but what is presented in the four panels remains elusive.

The legend of new Figure 6 has been updated to better explain the figure.

6. Fig. 1C: The text on the y-axis of the graph is 'Activity normalized to DDEI', but from the figure the activity appears to be normalized to DEEI. This should be corrected/clarified.

This has been corrected in the axis label of the new Figure 1C.

7. Manuscript text contains many typos and awkward sentences, and thus the text should be carefully edited.

The manuscript has been edited, and we hope the current text is clearer to the reader.

Reviewer #2:

1. In Fig. 5, reduced number of hair cell bundles in the NAA80 mutants is an important and interesting observation. Is this because the number of hair cells was reduced, or the number of stereocilia bundles per hair cell was reduced? When the number of stereocilia was reduced, how was the number/organization of kinocilia affected?

As shown in new Figure 6G-H, number of hair cells per crista were reduced, as well as the length of stereocilia. This has also been mentioned in the main text (line 229-230).

2. In Fig. 4, the results of mass spec should be strengthened if the authors can demonstrate that acetylation of other control proteins were not affected by the NAA80 mutation.

Our original conclusion was that Naa80 is necessary and sufficient to Nt-acetylate actin N-termini in zebrafish. We also find no evidence to support that Naa80 has any other *in vivo* substrates. To support this conclusion, we have added Supplemental Table S4, which contains all identified N-termini (peptide starting at position 1, 2 or 3) which were identified in both their Nt-acetylated and Nt-free (unacetylated) forms – 33 in total. We excluded N-termini which were only found to be Nt-acetylated and those which were wholly unacetylated because they would not be illuminating to this hypothesis. None of these N-termini (26 peptides) showed the pattern evident with the 7 actin N-termini, which is to be acetylated in fish with functional Naa80 (either +/+ or +/-), and unacetylated in *naa80* -/- fish. The following has been added to the manuscript (lines 191-198):

‘Other proteins may have been similarly affected by *naa80* knockout, showing Nt-Ac N-termini in the wild-type and *naa80* +/- samples and free N-termini in *naa80* -/- samples. We expected substrates of Naa80 to be Nt-acetylated in the wild-type and heterozygous samples and non-acetylated in the knockout samples. To test this, we compared the N-terminal peptides which had been found in both Nt-acetylated and Nt-free forms throughout the dataset (Supplemental Table S4). We found 33 N-termini with both acetylated and non-acetylated forms, including the 7 actin N-termini. Of these, actins were the only N-termini with acetylation status varying with *naa80* genotype. We thus found no evidence that proteins other than the actins were affected in their Nt-acetylation status by *naa80* knockout.’

Reviewer #3:

1. The authors identify the *naa80* gene as a zebrafish ortholog of human NAA80 and demonstrate the N-terminal acetylation activity of the Naa80 protein. However, it is not clear how the amino-acid sequence of the zebrafish Naa80 protein is similar to the amino-acid sequence of NAA80 proteins of other species including humans and mice. I request adding a figure to Figure 1 (or preparing a supplemental figure) and comparing amino-acid sequences between species.

We have aligned the Naa80 protein sequence with the mutant sequences in a new Supplemental Figure S2. These sequences are provided in Supplemental Table S5. A sequence alignment with different species' NAA80 orthologs has previously been published as a supplement to Ree et al., JBC 2020 (Fig. S2; [https://www.jbc.org/article/S0021-9258\(17\)50487-1/fulltext#supplementaryMaterial](https://www.jbc.org/article/S0021-9258(17)50487-1/fulltext#supplementaryMaterial)). In this paper we also discussed sequence differences between NAA80 orthologs.

2. The authors develop two *naa80*-null zebrafish alleles, 13del or 5del/1in, and report that *naa80*-null zebrafish may have difficulty in laying eggs. However, it is not clear how 13del and 5del/1in alleles disables the function of the *naa80* gene. I request adding a scheme to Figure 3 (or preparing a supplemental figure) to describe the modification(s) that the authors made to this gene, especially on a map of exons and introns. In addition, it is difficult to track how the authors crossed the zebrafish with the 13del or 5del/1in alleles and found the phenotype in laying eggs. I request preparing a supplemental figure illustrating the breeding described in the first and second paragraphs of the "*naa80* *-/-* zebrafish are free from gross morphological phenotypes and develop normally" section.

New Figure 3A shows how fish in each generation were genotyped and crossed to obtain hetero- and homozygous mutants. The conclusion regarding fertility was reached because we were able to obtain viable F3 larvae from incrossing *naa80* 13del *-/-* fish.

Regarding exons and introns, new Figure S1 A shows the exons of the *naa80* transcript. The transcript is also available in new Supplemental Table S5. The following text has been added to the manuscript (lines 98-102):

'There are two exons (137 and 930 bp) and one 3108 bp intron in this zebrafish gene. The two last bases of the first exon forms part of the start codon of the protein-coding region, which is almost completely contained in the longer second exon. Both the 5del/1in and 13del mutations are located within the second exon (Supplemental Figure S1 A).'

New Figure 3B and new Supplemental Figure S2 shows the effect of the mutant alleles on the predicted protein sequence.

3. The authors reported that *naa80*-null zebrafish display a reduced size of otolithic organ(s) in the inner ear, decreased number of bundles, weaker acoustic startle responses and reduced uptake of Yo-Pro-1. For these phenotypes, I request more details including (1) clarification of "bundles" in Figure 5E (are these kinocilia?) and (2) evaluation of stereocilia morphology (at least their length).

This has been clarified in new Figure 6G (previously Figure 5E), which counts 'total hair cells per crista'. It has also been stated in the main text (lines 229-230).

4. Please discuss the possible scenarios of how dysfunction of *naa80* causes in hair cells and otoliths. Do you think degradation of actin monomers is related to this phenotype? Do you think the speeds of polymerization and depolymerization are associated with the phenotypes?

We have added the following to the discussion section (lines 318-326):

'No evidence has been published that Nt-acetylation affects the half-life of actin monomers. One study in human WT and *NAA80* knockout cells also found that actin is not significantly modified by N-terminal arginylation and is not likely to regulate actin function and thereby the hearing loss

phenotype (Drazic et al., (2022) J Mol Biol). However, such *NAA80* knockout cells have been shown to have altered polymerization dynamics, altered cell morphology and increased speed of cell motility (Drazic et al., (2018) Proc Natl Acad Sci), as well as Golgi fragmentation (Beigl et al., (2020) Exp Cell Res). Specifically, unacetylated actin tends to both polymerize and depolymerize more slowly. These effects could all contribute to cell dysfunction and the specific phenotypes observed here. Focused studies of progenitor hair cells in the absence of *Naa80* could potentially elucidate the mechanism.'

Minor points:

Page 5, Figure 5E and 5F:

The titles of these two graphs don't seem to represent the data in the graphs. Is it possible to state like, "naa80 expression level during the development of zebrafish" and "naa80 expression levels in adult zebrafish tissues"?

This has been corrected in the revised Figure 2.

Page 6, "Male *naa80* 13del+/- fish were significantly shorter than controls ($p < 0.05$); however, *naa80* 13del-/- fish were not significantly shorter so the *naa80* genotype is not likely to be the cause of this.":

The term "short" may not be scientific. Is it possible to state like, "The body length of male *naa80* 13del+/- fish were (are)..."?

This has been rewritten to be more specific. The passage now reads (lines 153-155):

'Male *naa80* 13del+/- fish had an average body length of 3.447 cm, compared to 3.793 cm for wild-type fish which is a significant reduction ($p < 0.05$). However, *naa80* 13del-/- fish were 3.633 cm on average, suggesting that the *naa80* genotype is not likely to be the underlying cause.'

Page 8, the "Impaired N-terminal acetylation of cytoplasmic and muscle actins in *naa80* -/- zebrafish" section; Page 9, the "*naa80*-/- F0 knockout (KO) zebrafish displayed hearing-related defects" section; Page 10, Figure 5

In these sections and figures, it is uncertain which of the 13del-/- and 5del/1in-/- lines are used. Please clarify the lines used in these sections and figures. If both of these two lines are used, please separate the data.

In the first section mentioned (new Figure 5, previously Figure 4), the data is derived from proteomics analyses of adult tissues collected from stable mutant lines. Which mutant line is used is specified in the legend of the heatmap. For the transient CRISPR experiments in the second section mentioned, neither of the stable mutant lines were used. Eggs are injected at the 1-cell stage and larvae are used for the experiments directly, rather than establishing a stable line.

Page 10, "However, we did observe a reduction in otolith size compared to controls following Cas9 protein injection (Figure 5B)."; Page 10 Figure 5 legend:

The phrases, "reduction in otolith size" and "smaller otoliths", sound like the size of otoliths are small. I suspect that the authors intend to clarify that "the area covered by otolith" is small. Please rephrase if my understanding is correct. This is confusing since giant otoliths are observed in some diseases.

Since we indeed do intend to clarify that the otoliths are smaller in *naa80* KO larvae, we have refrained from rephrasing.

September 20, 2024

RE: Life Science Alliance Manuscript #LSA-2024-02795-TR

Dr. Rasmus Ree
NORCE Norwegian Research Centre
Thormøhlens gate 55
Bergen 5006
Norway

Dear Dr. Ree,

Thank you for submitting your revised manuscript entitled "Naa80 is required for actin N-terminal acetylation and normal hearing in zebrafish". We would be happy to publish your paper in Life Science Alliance pending final revisions necessary to meet our formatting guidelines.

- please be sure that the authorship listing and order is correct
- please add the Twitter handle of your host institute/organization as well as your own or/and one of the authors in our system
- please add the author contributions, a separate conflict of interest statement, and the figure legends to the main manuscript text. Please consult our manuscript preparation guidelines <https://www.life-science-alliance.org/manuscript-prep> and make sure your manuscript sections are in the correct order
- please use the [10 author names, et al.] format in your references (i.e. limit the author names to the first 10)
- please add a figure callout for Figure 1A, Figure S1B,C; please double-check your figure callouts for your Figure 5 and 6. It seems like you labeled Figure 5 panels when you mean to refer to Figure 6. Please make sure to add a figure callout for each panel for Figure 6.
- the dataset uploaded to PRIDE should be made publicly accessible at this point, removing the need for the Reviewer access information in the Data Availability statement

LSA now encourages authors to provide a 30-60 second video where the study is briefly explained. We will use these videos on social media to promote the published paper and the presenting author (for examples, see <https://docs.google.com/document/d/1-UWCfbE4pGcDdcgzcmiuJl2XMBJnxKYeqRvLLrLS08s/edit?usp=sharing>). Corresponding or first-authors are welcome to submit the video. Please submit only one video per manuscript. The video can be emailed to contact@life-science-alliance.org

A. FINAL FILES:

B. MANUSCRIPT ORGANIZATION AND FORMATTING:

Sincerely,

September 27, 2024

RE: Life Science Alliance Manuscript #LSA-2024-02795-TRR

Dr. Rasmus Ree
NORCE Norwegian Research Centre
Thormøhlens gate 55
Bergen 5006
Norway

Dear Dr. Ree,

Thank you for submitting your Research Article entitled "Naa80 is required for actin N-terminal acetylation and normal hearing in zebrafish". It is a pleasure to let you know that your manuscript is now accepted for publication in Life Science Alliance. Congratulations on this interesting work.

DISTRIBUTION OF MATERIALS:

Again, congratulations on a very nice paper. I hope you found the review process to be constructive and are pleased with how the manuscript was handled editorially. We look forward to future exciting submissions from your lab.

Sincerely,
